# Phase Space Reconstruction from a Biological Time Series: A Photoplethysmographic Signal Case Study

**Javier de Pedro-Carracedo** [1,2] **, David Fuentes-Jimenez** [3]**, Ana María Ugena** [4] **and Ana Pilar Gonzalez-Marcos** [1,*]

1   Photonic Technology and Bioengineering Department, Technical University of Madrid (UPM), 28040 Madrid, Spain; javier.depedro@uah.es
2   Computer Engineering Department, University of Alcalá (UAH), Alcalá de Henares, 28871 Madrid, Spain
3   Department of Electronics, University of Alcalá (UAH), Alcalá de Henares, 28871 Madrid, Spain; david.fuentes@depeca.uah.es
4   Departamento de Matemática aplicada a las Tecnologías de la Información, Technical University of Madrid (UPM), 28040 Madrid, Spain; anamaria.ugena@upm.es
*   Correspondence: anapilar.gonzalez@upm.es

**Abstract:** In the analysis of biological time series, the state space is comprised of a framework for the study of systems with presumably deterministic and stationary properties. However, a physiological experiment typically captures an observable that characterizes the temporal response of the physiological system under study; the dynamic variables that make up the state of the system at any time are not available. Only from the acquired observations should state vectors be reconstructed to emulate the different states of the underlying system. This is what is known as the reconstruction of the state space, called the phase space in real-world signals, in many cases satisfactorily resolved using the method of delays. Each state vector consists of $m$ components, extracted from successive observations delayed a time $\tau$. The morphology of the geometric structure described by the state vectors, as well as their properties depends on the chosen parameters $\tau$ and $m$. The real dynamics of the system under study is subject to the correct determination of the parameters $\tau$ and $m$. Only in this way can be deduced features have true physical meaning, revealing aspects that reliably identify the dynamic complexity of the physiological system. The biological signal presented in this work, as a case study, is the photoplethysmographic (PPG) signal. We find that $m$ is five for all the subjects analyzed and that $\tau$ depends on the time interval in which it is evaluated. The Hénon map and the Lorenz flow are used to facilitate a more intuitive understanding of the applied techniques.

**Keywords:** biological signal; PPG signal; phase space reconstruction; method of delays

## 1. Introduction

Dynamic systems, as could be any physiological system, are mathematically characterized by differential equations. The identification of the simplest physiological model, to describe the physiological temporal evolution, requires specifying the minimum dimension of the dynamic system, that is the number of dynamic variables or differential equations involved in the evolution of the system. Our first objective is to describe the methodology, most widely used in general nonlinear dynamical systems, but not in biological signals, and based on the lag or delay method, to obtain the minimum number of dynamic variables of the system (human body) that generates a concrete biological signal in a physiological state. As a second objective, we apply this methodology to the PPG signal. Our results indicate a fifth dimension, one more than that obtained by [1] using the method of embedding to reconstruct the attractor from experimental biological data.

The dynamic variables compose the components of a state vector in the state space: the state space is comprised of a coordinate system with as many coordinates as dynamic variables the system presents and allows embracing the set of all possible states of the system, which describes the dynamics of the system. At each time instant, the state vector is in a different position (an isolated point in the state space); the chronological evolution of these points draws a trajectory in the state space. When the trajectory extends to infinity, it is known as an orbit [2].

The time evolution of each dynamic variable involves the most natural way to characterize any dynamic system; this is nothing but the usual representation of the value of each dynamic variable as a function of time [3]. Another way to describe a dynamical system graphically is to replace the time-independent variable, typically the variable time *t*, with another dynamic variable of the system [4,5]. In this case, each point on the graph, in a two-dimensional coordinate system, represents the system state in a given time instant; with a third dynamic variable, a three-dimensional coordinate system is available, the highest possible visual capacity. In a hypothetical abstract space, the coordinate system must cover all dynamic variables of the system under study. Each point of the theoretical graph represents a system state at a given time instant. This graphical representation designates what is called the state space, where each point denotes a state vector. If the dynamic variables that make up the state vectors evolve according to established physical laws (primitive concepts), as real-world signals, i.e., biological signals, the state space becomes a space restricted to states with physical significance; it is from now on the so-called phase space. The graphical representation of different trajectories traced by the state vectors in the phase space is termed the phase diagram.

The graphical analysis of a dynamic system is reduced to a state space of at most three-dimensions, implicitly including the time variable [6], without forgetting that other unknown variables, apart from the three considered, can be significant in the system dynamics. Since most real-world physical systems are nonlinear with an ever-present coupled noise, from a graphical analysis with the appropriate dynamic variables, the underlying dynamic system deterministic structure can be deducible, not apparent in a possible erratic evolution in the time domain. A more complex study, although less visual, with many more variables, requires a mathematical formalism not yet fully consolidated [7].

In physiological terms, typically, in a clinical trial, it is customary to acquire a unique biological signal identifying each physiological response, so that not all the dynamic variables involved in the time evolution of the system under study are available. Thus, each biological signal represents a response of a physiological system; each response distinguishes the time evolution of a dynamic variable. Accordingly, from the scalar measurements that define the acquired biological signal, the different state vectors must be generated in order to recreate the different states that characterize the physiological system dynamics in question. The phase space reconstruction allows faithfully reproducing the evolution of the system under study from a simple biological signal, in the absence of the dynamic variables involved.

Usually, observations or measures at regular time intervals, i.e., one every $\Delta t$ seconds, are an example of what in academic jargon is know as a time series. The most common method to define each state vector is to use delayed versions of the observations as state vector components. In the simplest case, in a two-dimensional coordinate system, as illustrated by Figure 1 for the two representative dynamical systems that we use in this paper to explain the methodology employed, the ordinates' axis represents the measurement value $x_n$ at time $n\Delta t$ and the abscissas' axis the measurement value $x_{n-1}$ at time $(n-1)\Delta t$. Hence, each point in the plot identifies the $\mathbf{x}_n = (x_{n-1}, x_n)$ state vector.

At the most general level, the number *m* of components of the state vector and the delay $\tau$ between components are variable, and this depends on the observable of the physical system being measured, that is $\mathbf{x}_n = (x_{n-(m-1)\tau}, \ldots, x_{n-\tau}, x_n)$. From a sequence of scalar measurements, it can reconstruct every state vector, in *m*-dimensional phase space, following the lag or delay method [8]. The state space reconstruction quality depends on the proper estimation of the parameters $\tau$ and *m*, considering the time delay embedding theorem [9–11], which provides the ideal conditions under which to reconstitute a chaotic dynamic system from a sequence of observations of the state of the dynamic system.

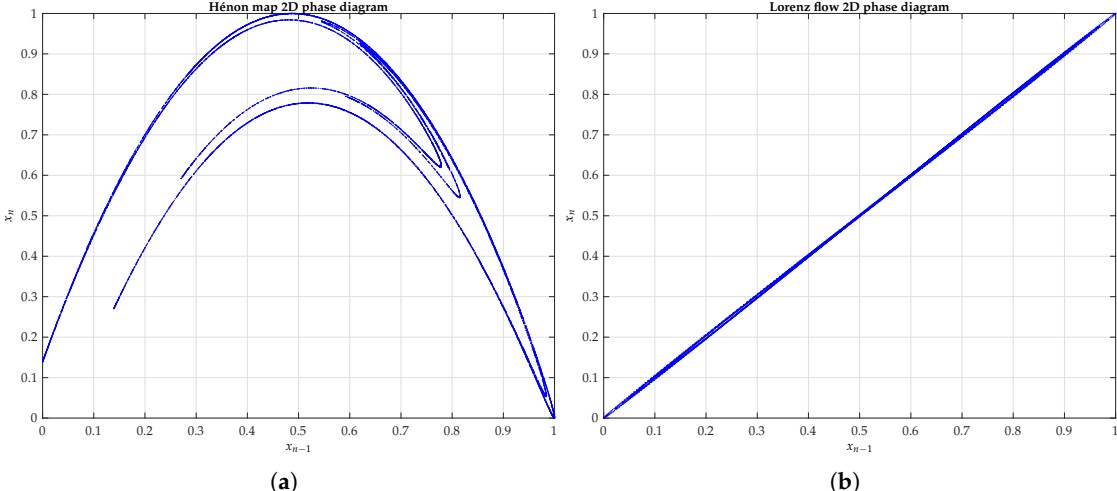

**Figure 1.** 2D phase diagram from the dynamic variable $x$ of the: (**a**) Hénon map; and (**b**) Lorenz flow, with a sampling time of 2 ms.

State or phase space reconstruction, in contrast to traditional signal analysis in the time or frequency domain, constitutes the cornerstone of the time series analysis in terms of nonlinear dynamics [12,13]. It is the first step in the analysis process of the real functional nature that reflects the physical system in question. In doing so, it should pay special attention to determine the optimum values of $\tau$ and $m$ [14] efficiently, so that state space reconstruction faithfully allows characterizing the dynamics of the original system. We detail the most common techniques, elegant in their simplicity, which enable a first approximation of the state space reconstruction and then apply them to a particular biological signal, the PPG signal, extracting interesting conclusions that lead to promising future studies.

In a typical trajectory, over the phase space, dynamic variables' time evolution can tend to infinity or be confined to a bounded region of the phase space. If the dynamic system is dissipative, once the transient response finishes, its dynamics will tend to a subset of the state space called the attractor. This subset is invariant under the dynamical system's evolution. In a chaotic system, the attractors describe very complex geometric objects, having a typical fractal structure; they are the so-called strange attractors [15].

The attractor geometry provides valuable information not only on the dynamic nature (Lyapunov exponents) of the underlying physical system, but also about the structural complexity sustaining that dynamics (dimensionality), hence the vital importance of a successful phase space reconstruction. The connections between these factors go beyond the scope of this paper, though we will deal properly with these issues in future communications.

In this paper, we outline the due process to be followed in the phase space reconstruction from one scalar time series, applying the methodology to a relatively well-known and easily accessible biological signal, the PPG signal (see Figure 2b). In Section 2, we briefly describe the basic characteristics of the biological signal used in this work, the PPG signal, just like the main mechanisms underpinning the state space reconstruction, focusing mainly on how to determine the reconstruction parameters correctly, the lag $\tau$, and the embedding dimension $m$. As such, it is essential to evaluate previously the degree of determinism and stationarity present in the time series, in order to give significance to the obtained results. In order to do this, we introduce several graphic methods to assess the approximate determinism present intuitively in the dynamic system evolution, making the state space reconstruction meaningful. In addition to this, we corroborate the determinism and stationary nature of biological signals by formal procedures (see Figure 2a). In Section 3, we show the obtained results in line with the methodology referred to above. Finally, in Section 4, we analyze and interpret the obtained results laying the ground for forthcoming works.

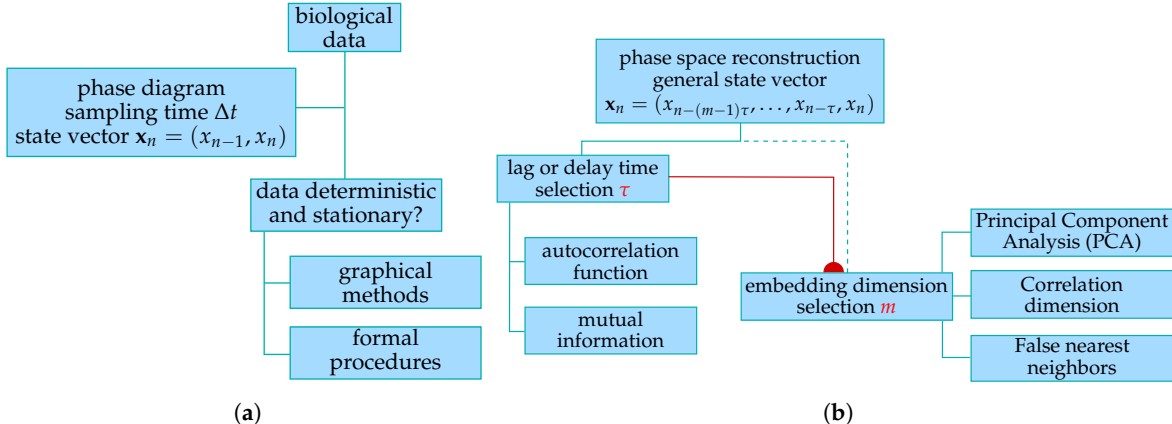

**Figure 2.** From a sequence of scalar measurements: (**a**) data and requirements; and (**b**) the reconstruction done in this paper for the PPG signal following the lag or delay method. The description of the concepts is in Section 2.2, and the application results are in Section 3.

## 2. Materials and Methods

### 2.1. Biological Data

Health monitoring by non-invasive means has attracted the attention of medical specialists for some years now since it allows advancing preliminary diagnoses on possible pathological dysfunctions, whether chronic or transient. Some scientific evidence sustains the relationship between the biological signals generated by the human body and the health status of the individual, as skin temperature (ST), electrodermal activity (EDA), pulse wave or photoplethysmography (PPG), electrocardiography (ECG), electromyography (EMG), respiration (Resp.), pupil diameter (PD), electroencephalography (EEG), and blood pressure, among others. The information provided by all these biological signals depends on the knowledge available about the underlying physiological processes.

The study of the dynamics of all these biological signals will help to understand better the physiological system that generates them. Furthermore, it would help to get an insight into how different dynamic variables couple, as, for instance, the heart and respiratory frequencies, in order to keep the physiological system in a perfect condition, preventing its deterioration towards a pathological state. In this paper, we focus only on a single biological signal, the PPG signal; future publications will describe the results for more biological signals.

We chose the PPG signal because it is easily accessible. A pulse oximeter consists of a light emitter and a photodetector that collects and records (pulse or PPG signal) the loss—scattering and absorption—that a beam of light undergoes when it passes through, or is reflected by, a human tissue [16]. It allows detecting blood volume changes in the microvascular bed of tissue—in our case, the middle finger of the left-hand—obtaining valuable information about the cardiovascular system and, on the whole, about the cardiorespiratory system. Given the simplicity of its non-invasive accommodation, in addition to its low cost, a pulse oximeter is very useful in biomedical applications for clinical [17] and sports environments [18]. As with other biological signals, characteristics extracted from the PPG signal allow to no small extent identifying ideal health conditions and their possible deviations. Thus, indicators associated with different pathologies could be established, which anticipate their severity according to the causes that gave rise them. Typically, these indicators in the case of PPG signal were based on the morphology of the signal rather than on its dynamics [17,19]; we think that by studying dynamic aspects of the PPG signal, the physiological system that generates it can be better understood.

With the aim of examining the PPG signal dynamics, we used the PPG signals (to show in this paper only five individuals chosen randomly) from a total of 40 students, between 18 and 30 years old, and non-regular consumers of psychotropic substances, alcohol, or tobacco, selected to participate in a

national research study [20,21]; the five PPG signals shown in this paper were the same as those used in previous research that confirmed the predominantly quasi-periodic behavior for small timescales in healthy young people with a modified version of the 0–1 test [22]. Showing more than five subjects will not clarify the proposed method. The results would be similar. Remember that the fundamental frequency of the PPG signal is typically around 1 Hz, depending on the heart rate (0.5–4 Hz, first and second harmonic) and respiratory activity at roughly 0.2–0.35 Hz. We applied a Butterworth bandpass filter with cutoff frequencies at 0.01 and 8 Hz, in order to avoid high-frequency noise and, to some extent, motion artifacts [23]. All signals were captured from the middle finger of the left hand and sampled at a frequency of 250 Hz [20,21], say sampling time $\Delta t = 4$ ms.

### 2.2. Determinism and State Space Representation

From a linear perspective, the erratic (or irregular) behavior of the one system response is due to a random external component. The chaos theory finds that a random input is not the only cause to get an irregular output. A simple deterministic equation, as with nonlinear chaotic systems, can generate irregular data without the contribution of any external input.

In a deterministic system, according to the deployed dynamical systems theory [24], once having known its current state, the future states are entirely determined. The system dynamical and geometric properties all are to some extent included in the state space representation. Equations of motion that describe the behavior of a physical system as a function of time are deployed in the state space. Henceforth, the drawn trajectories in the state space represent the time evolution that the state vector is undergoing over time. For this reason, the state space or phase space can be used to get an approach to the rules that govern the dynamical system evolution.

In an experimental setting, the system equations are not available. Usually, this is obtained solely with one system response projection in a one-dimensional space, a scalar time series, which represents the acquired measurements of the system in question. The state space reconstruction method aims to reconstruct the state vectors from the time series, so that the time evolution of these vectors replicates a dynamics equivalent to that of the original system, as shown in Figure 3. A time series reflects roughly the time evolution of a single dynamic variable, but as J.Doyne Farmer once said, the evolution of a variable depends on other variables of the system or, in other words, its value is within the history of other variables with which it interacts [25].

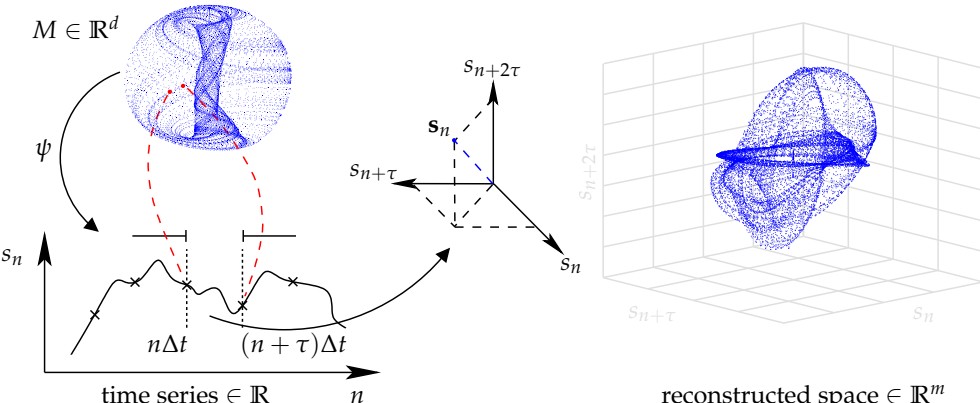

**Figure 3.** State space reconstruction general overview.

Ideally, a first-order ordinary differential equation system taking action on the state space [2,12,26] defines a dynamical system. A set of possibly infinite states and certain transition rules that specify how the system moves from one state to another can describe several systems. For a finite-dimensional state space $\mathbb{R}^m$, a state is defined by a vector $\mathbf{x} \in \mathbb{R}^m$. Henceforth, the dynamics of the system can be described by an $m$-dimensional map or by an $m$ first-order ordinary differential equation system called flow. In the first instance, such as the Hénon map, the time is a discrete variable,

$$\mathbf{x}_{n+1} = \mathbf{F}(\mathbf{x}_n), \quad n \in \mathbb{Z}, \tag{1}$$

while in the latter, as is the Lorenz flow, the time is a continuous variable,

$$\frac{\mathrm{d}}{\mathrm{d}t}\mathbf{x}(t) = \mathbf{f}(\mathbf{x}(t)), \quad t \in \mathbb{R}. \tag{2}$$

For each initial condition $\mathbf{x}_0$, or $\mathbf{x}(0)$, the solutions to Equations (1) and (2), a sequence of points $\mathbf{x}_n$, o$\mathbf{x}(t)$, respectively, describe a dynamic system trajectory. With its time evolution, a typical trajectory can tend to infinity or be confined to a bounded region in the state space. All the initial conditions that lead to the same asymptotic behavior of the observed trajectory are known as the basin of attraction [27].

The trajectory described in the state space, from a single observable, is a presumable indication of the presence of a deterministic behavior if the states are not arranged as a point cloud, despite an erratic appearance in the time domain. In a 2D phase diagram, the variable time, on the abscissa axis, is replaced by the observable value in a prior time determined by the parameter $\tau$, as discussed in Section 1; in a 3D phase diagram, except for the first axis, all other axes pick up previous values of the observable variable separated from each other a time $\tau\Delta t$.

### 2.2.1. Graphical Methods for Assessing the Presence of Determinism

The following items summarize two methods of visualizing a possible hidden determinism.

a.  Next-amplitude plot: From successive maximums detected in time series, each one $M_n$ is represented (abscissa) versus its immediate or subsequent successor $\tau$ times $M_{n+\tau}$ (ordinate) ahead. A well-defined curve, like the one in Figure 4a, could reveal the presence of chaos, although the noise could mask a correct interpretation.
b.  Difference plot: The graphic's coordinates are delayed differences between successive observations, whether immediate or separated $\tau$ number of times. On the abscissa axis, it is represented as $\Delta s_n = s_{n+1} - s_n$, and on the ordinate axis, the next difference $\Delta s_{n+\tau} = s_{n+\tau+1} - s_{n+\tau}$. In the simplest form, the first difference plot, with a delay $\tau = 1$, on the abscissa is represented as $s_{n+1} - s_n$ and on the ordinate $s_{n+2} - s_{n+1}$. The presence of an infinitely continuous curve, as illustrated in Figure 4b, evidences a high degree of underlying determinism.

### 2.2.2. Formal Procedures for Assessing the Presence of Determinism and Stationarity

To assess the presence of determinism in PPG signal segments of a significant size, we first selected stationary fragments of the PPG records for analysis. These fragments were selected using an approach based on the second-order weak stationarity, i.e., constant mean and variance with the autocorrelation depending only on the time difference. To do this, we partitioned the data into time bins consisting of a PPG cycle. Next, the mean and variance for each bin were calculated, and then, we looked for consecutive time bins in which these values did not change significantly from those calculated for the entire time series, discarding the remainder of not statistically significant time bins. For the selected intervals, it was checked if the autocorrelation function depended on the time difference alone or not. This allowed us to ensure that the fragments of the PPG signal were representative of the dynamics of the entire time series.

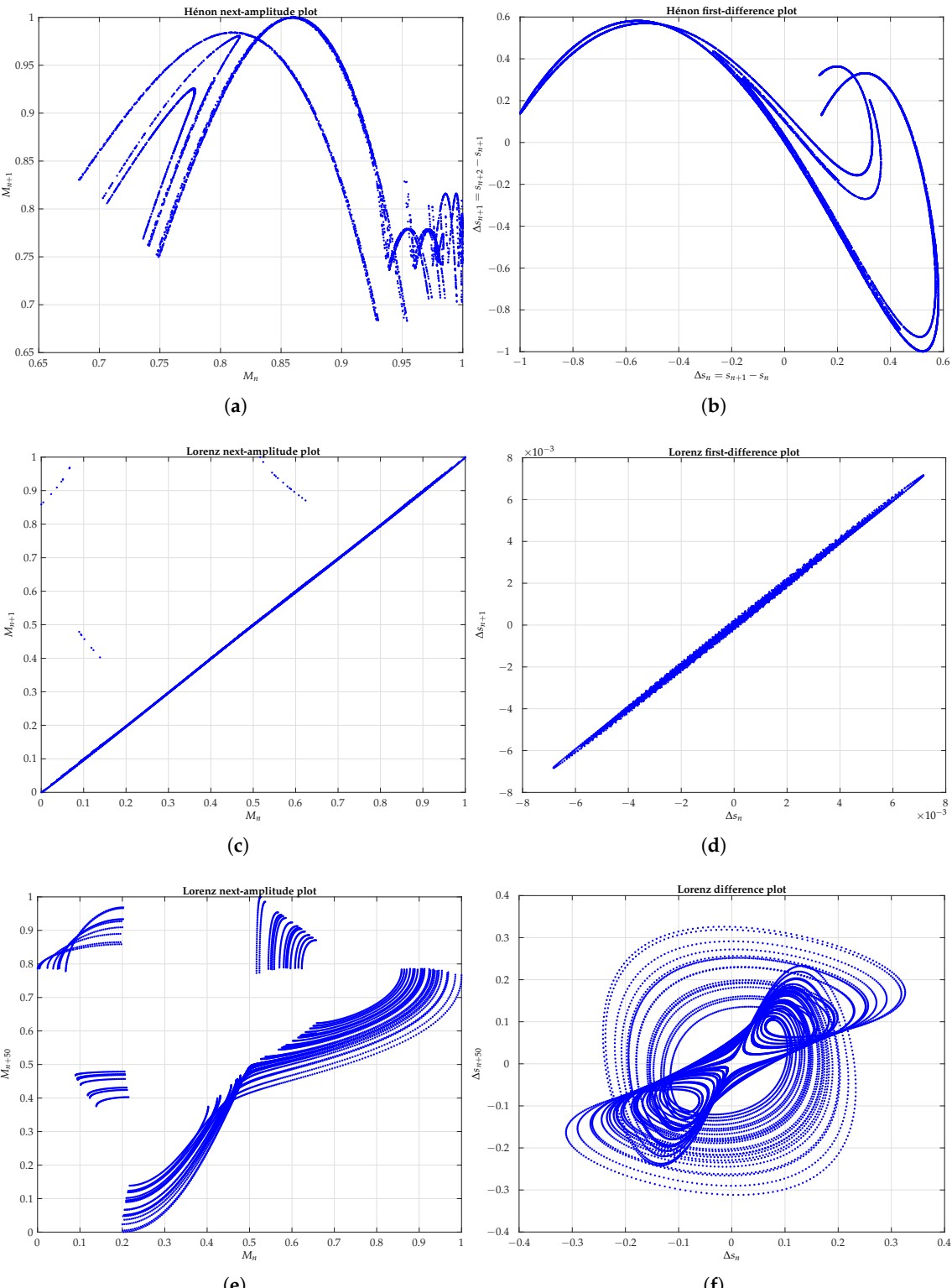

**Figure 4.** Next-amplitude and difference plots, from the dynamic variable *x* of the Hénon map and the Lorenz flow with a sampling time of 2 ms: (**a**,**b**) Hénon map with $\tau = 1$; (**c**,**d**) Lorenz flow with $\tau = 1$; (**e**,**f**) Lorenz flow with $\tau = 50$.

In order to examine the determinism present in the PPG data, we used the central tendency measure (CTM), as it provides a quantitative estimate for the smoothness of the attractor's trajectory. The index of smoothness $I_s$ of the trajectory, defined as a ratio of the CTM of the angle variations

of the successive tangent vectors for the original time series to that for its surrogate data, is an effective measure for the smoothness of the trajectory and, thus, for determinism in the time series [28]. The larger the index of smoothness $I_s$ of the trajectory, the less smooth is the attractor's trajectory, hence a reduced determinism presence. Since the surrogate data are linear stochastic processes with a power spectrum identical to that of the original data (raw time series), the method of surrogate data makes it possible to detect nonlinear determinism on a statistical basis [29,30].

### 2.3. Phase Space Reconstruction Method

It was in 1980 that Packard et al. [31] first tackled the problem of how to connect the phase space or state space vector $\mathbf{x}(t)$ of the dynamical variables from one physical system to a possible time series $\{s_n\}$ measured in any experiment.

A time series is a sequence of scalar measures $s_n$ of an observable, acquired at regular times $\Delta t$. The different measures depending on the current state of the system,

$$s_n = \psi(\mathbf{x}(n\Delta t)) + \eta_n, \tag{3}$$

where $\psi$ represents an observable measurement function and $\eta_n$ the measurement noise, which characterizes the random nature of the imprecision of the measure [32].

An $m$-dimensional reconstruction of the $\mathbf{s}_n$ state vectors, from scalar measurements, is given by:

$$\mathbf{s}_n = (s_{n-(m-1)\tau}, \ldots, s_{n-\tau}, s_n), \quad \tau \in \mathbb{Z}^+. \tag{4}$$

The time interval between adjacent coordinates of the state vector is $\tau\Delta t$, and it is known as the lag or delay time. Formally, it has been shown that if $m$ is higher than two times the capacity dimension of the attractor ($\dim_C$), a one-to-one correspondence between the reconstructed attractor and the real attractor is guaranteed, no matter how large the dimension of the original state space is [10,11].

### 2.3.1. Lag or Delay Time Selection

In theory, with an unlimited number of measurements without noise, any delay is equally valid. In practice, with real experimental data, the geometrical and dynamic properties that characterize the attractors differ from each other, depending on the delay value chosen [26]. If the sampling time $\Delta t$ is tiny, the values of consecutive samples are very similar, $s_n \approx s_{n+\tau}$, and with $\tau = 1$, the represented points contain much redundant information. Furthermore, with a small delay, the noise level can blur or hide any local geometric structure.

Regardless of the form that the attractor can take, the graphic arrangement of perfectly related pairs of points describes a straight line, as an identity function (a diagonal line of 45°) without any meaning, as illustrated in Figure 5a.

In order for the distribution of points to provide some geometric significance, the values of the components that make up the coordinate system must be sufficiently independent. In the real state space, in which the coordinates correspond to different variables, this condition is satisfied. In the same vein, a delay too large can be counterproductive. With a significant delay, the dynamic relationship between the values of the variable disappears, so that, as the delay value increases, the geometric structure of the points becomes more complex and diffuse until finally the points are dispersed randomly on the state space (see Figure 5b).

In Figure 6, the Lorenz flow, with the correct delay time $\tau = 16$, is represented. The next points describe how to calculate it.

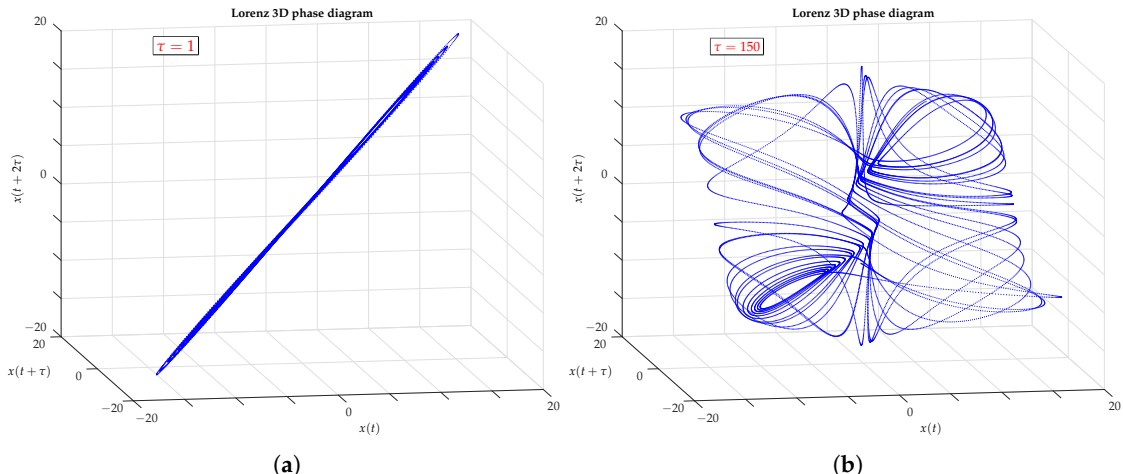

**Figure 5.** State space reconstruction from the dynamic variable $x$ of the Lorenz attractor, with a sampling time $\Delta t$ of 2 ms: (**a**) with $\tau = 1$; and (**b**) with $\tau = 150$.

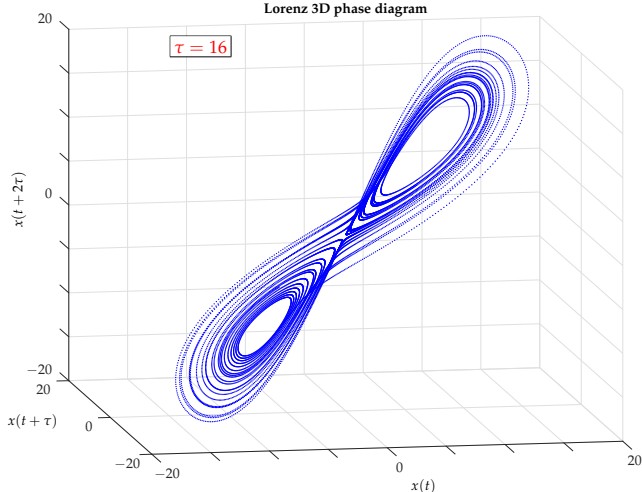

**Figure 6.** State space reconstruction from the dynamic variable $x$ of the Lorenz attractor, with a sampling time $\Delta t$ of 2 ms and $\tau = 16$.

Autocorrelation Coefficients

The autocorrelation coefficients $R_\tau$ measure the correlation degree of a variable with itself at different instances. Its constituents are autocovariance and variance.

$$
\begin{aligned}
R_\tau &= \frac{\text{autocovariance}}{\text{variance}} \\
&= \frac{\frac{1}{N} \sum_{i=1}^{N-\tau} \left(s_i - \langle s \rangle\right) \left(s_{i+\tau} - \langle s \rangle\right)}{\frac{1}{N} \sum_{i=1}^{N} \left(s_i - \langle s \rangle\right)^2},
\end{aligned}
\tag{5}
$$

where $N$ represents the time series length and $\langle s \rangle$ the arithmetic mean of all the time series observations.

According to Equation (5), with a minimum delay $\tau = 0$, the coordinates of each point are identical ($s_n = s_{n+\tau}$), and the autocorrelation $R_{\tau=0} = 1$ is maximal. As delay increases, autocorrelation decreases until it eventually is reduced to zero, or as happens in real data, there is a fluctuation around zero, within a narrow margin, due to the noise in the data. In short, the autocorrelation coefficients range from $+1$ y $-1$. Maximum values, $R_\tau = \pm 1$, are perfectly correlated data; minimum values, $R_\tau = 0$, correspond to uncorrelated data.

Autocorrelation coefficients, for successive delays, conform to the autocorrelation function. It is also known as the spectral autocorrelation coefficient [33] (it is recommended to use $\lfloor N/4 \rfloor$ delays with a time series of $N > 50$ observations). The graphical representation of $R_\tau$ is known as the correlogram. To a certain extent, a correlogram shows the type of regularity in the data. In the case of trendless and uncorrelated data, 95% of the autocorrelation coefficients are contained, in theory, in a band of $\pm 2/\sqrt{N}$, around zero. About 5% of them can exceed the indicated limit without losing its condition of uncorrelated data [34].

From the autocorrelation function, it is possible to adopt some criteria for selecting the optimal delay. Among the various selection criteria, the most commonly used determines as the optimal delay the first zero crossing of the autocorrelation function, as illustrated in Figure 7a. As an example of an alternative approach, in climate related attractors [35], other criteria apply the one known as correlation time (when $R_\tau$ decreases to a value of $1/e \approx 0.37$).

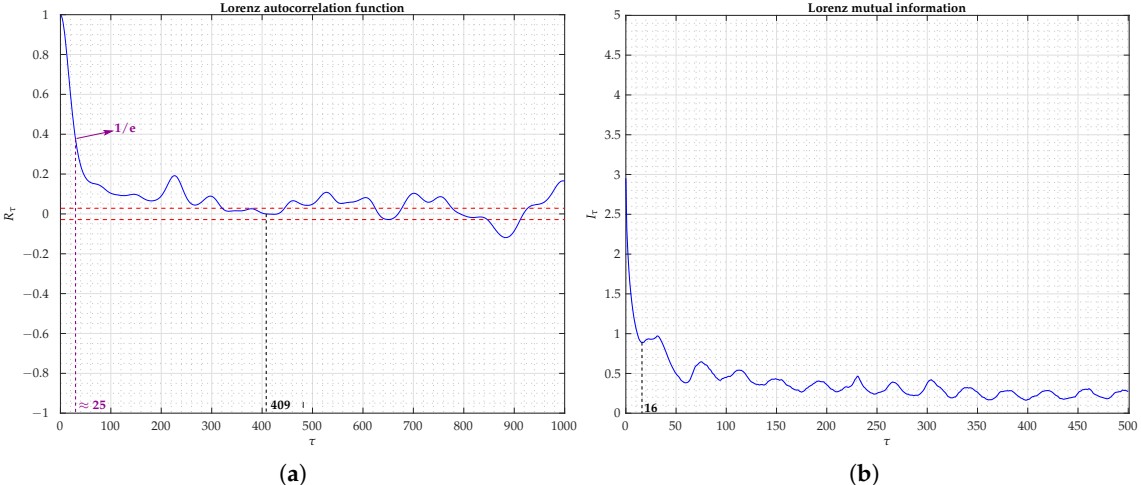

**Figure 7.** From the dynamic variable $x$ of the Lorenz attractor: (**a**) autocorrelation function (AF), where the first zero crossing is when $\tau = 409$; and (**b**) mutual information (MI), where the first minimum is with $\tau = 16$.

Mutual Information

Autocorrelation coefficients consider the degree of the mutual relationship on a linear basis, inappropriate in nonlinear systems [14]. In these situations, it is best to use the mutual information, as it enables determining, on a probabilistic basis, to what extent two values of the same variable, measured at different time instants, relate to each other. For example, if the point coordinates are identical, with $\tau = 0$, these represent the same information, and therefore, one coordinate accurately predicts the other or, otherwise, the amount of information that one coordinate contains about the other, mutual information, is maximal.

Whenever there is any relationship between two values of the same variable, acquired at different time instants, one value contains information about the other value. That is, one value helps predict the other value or, in other words, the knowledge of one value reduces the uncertainty of the other value. The uncertainty reduction is called mutual information. If **X** denotes a time series, $\{s_n\}$, ergo, $s_1, s_2, \ldots, s_N$, with $N$ observations, and **Y** the delayed time series, $\{s_{n+\tau}\}$, ergo, $s_{1+\tau}, s_{2+\tau}, \ldots, s_N$, with $N - \tau$ observations, where $\tau$ indicates the delay; the average mutual information $I_{\mathbf{Y};\mathbf{X}}$ between both time series can be expressed, in probabilistic terms, as:

$$
\begin{aligned}
I_{\mathbf{Y};\mathbf{X}} = I_\tau &= \sum_{i=1}^{N_c} \sum_{j=1}^{N_c} P(x_i, y_j) \log \frac{P(x_i, y_j)}{P(x_i) P(y_j)} \\
&= \sum_{i=1}^{N_r} P(s_i, s_{i+\tau}) \log \frac{P(s_i, s_{i+\tau})}{P(s_i) P(s_{i+\tau})},
\end{aligned} \tag{6}
$$

where $N_c$ is the number of cells containing points, with non-zero probability, and $N_r$ is the number of routes, $s_i s_{i+\tau}$, in the state space. Equation (6), in terms of entropy, is rewritten as:

$$
I_{\mathbf{Y};\mathbf{X}} = H_{\mathbf{Y}} + H_{\mathbf{X}} - H_{\mathbf{X},\mathbf{Y}}, \tag{7}
$$

where $H_{\mathbf{X}}$ is the entropy of $\mathbf{X}$, $H_{\mathbf{Y}}$ the entropy of $\mathbf{Y}$, and $H_{\mathbf{X},\mathbf{Y}}$ the joint entropy of $\mathbf{X}$ and $\mathbf{Y}$. Therefore, somehow, the mutual information involves a measure of predictability of the system, that is a measure of the degree of knowledge of $s_{n+\tau}$ denoted $s_n$.

In order to reconstruct an attractor, as faithfully as possible, the minimum mutual information and delay value are required. As with the autocorrelation function, the mutual information decreases as the delay increases, until it eventually becomes zero. The first minimum of the mutual information includes a possible criterion for selecting the optima delay value [8], as shown in Figure 7b. One limitation of this method is that many points are needed if we are to get a consistent result.

### 2.3.2. Embedding Dimension Selection

There is no rule of thumb to set the minimum reconstruction dimension $m$, and none of the published proposals is widely accepted, so it is a good idea to use more than one method on the same data. Among all possible techniques, the correlation dimension and false nearest neighbors stand out, though principal components analysis, as a preliminary attempt, may shed some light.

From a scalar time series $\{s_n\}_{n=1}^N$ acquired and with the proposed $\tau$ value as stated above, $N - (m-1)\tau$ vectors with $m$ components per vector are defined, where each component symbolizes an alleged dynamic variable of the physical system. To a certain extent, we initiate a multivariate analysis with $m$ time series data obtained by Equation (4).

### Principal Component Analysis

A first approach to the estimation of the parameter $m$ comes from principal component analysis (PCA), an application of linear algebra.

Figure 8a shows the results for the Lorenz attractor. This method can help to extract relevant information from seemingly complex data, a priori reducing the number of dimensions needed to characterize the dynamics present in the data, that is to identify the basis that best represents a noisy dataset, detecting those redundant dimensions that record the same dynamic information [13]. This basis, to a certain extent, filters the noise and improves the recovery of the hidden dynamics in the data. In all, this linear composition aims to find the smallest subspace (hyperplane) that contains roughly the attractor.

The PCA method takes up an $m_i$-dimensional coordinate system ($m_i$ spatial directions) much higher than the $m$-dimensional reconstruction space that supposedly characterizes the attractor. The experimenter's intuition plays an important role in deciding on a greater or lesser initial value of $m_i$. This method recognizes the directions of the $m_i$-dimensional coordinate system that shows more significant variations (variances) in the data, the $m$-dimensional reconstruction space, discarding those other directions in which small variations may come from fluctuations of the coupled noise on data. Discarding those less important directions, apart from reducing the effect of noise, particularly white noise, makes it possible for the appearance of a more simplified dynamics from an originally very high-dimensional space. Hence, it enables a dimensional reduction. The main disadvantage of this method is its subjective nature when determining the final value of $m < m_i$, provided that the

differences between the principal directions and those directions attributed to noise are not clearly significant. Typically, with real data, or even with an unfortunate initial choice of $\tau$ and $m_i$, the differences between the variances of the different directions $m_i$ are not always so evident, and it is necessary to define a somewhat arbitrary threshold that classifies the directions as primary ($m$ principal components) or secondary (coupled noise). Furthermore, there is no guarantee that the reconstruction will always be optimal, since the reconstruction method relies on a non-parametric analysis, and sometimes, it is unable to distinguish between a chaotic signal and the noise itself when they have a similar power spectrum [36].

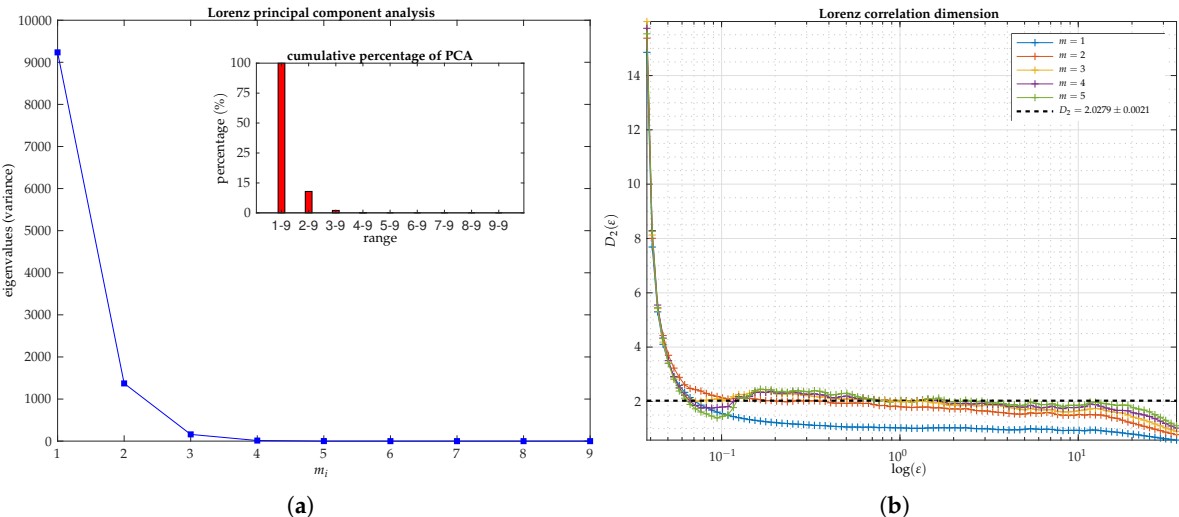

(a)        (b)

**Figure 8.** From the dynamic variable $x$ of the Lorenz attractor: (**a**) principal component analysis (PCA). The inner box clearly shows how the dynamic behavior of the attractor can be characterized with only three directions. (**b**) Correlation dimension ($D_2$). It can be seen how from $m = 3$, the correlation dimension is confined to a constant value in an intermediate and reduced range of scales $\varepsilon$.

Correlation Dimension

The correlation dimension involves the most usual measure of dimension, mainly because of its computational efficiency. It is based on spatial correlation [37]. The general procedure includes point counting within a distance $\varepsilon$, evaluated for each of the points that make up the state space. The normalized total number of points, for a particular distance $\varepsilon$, is called the correlation sum, the estimator of the correlation integral,

$$C_\varepsilon = \frac{\text{points within a distance } \varepsilon}{N(N-1)}, \tag{8}$$

where $N$ is the time series length. For large enough $N$,

$$C_\varepsilon = \lim_{N \to \infty} \frac{\text{points within a distance } \varepsilon}{N^2}, \tag{9}$$

which applies for an $N$ value of several hundred data; according to the mathematical formalism,

$$C_\varepsilon = \lim_{N \to \infty} \frac{1}{N^2} \sum_{i=1}^{N} \sum_{j=1}^{N} \Theta \left( \varepsilon - \left\| \mathbf{s}_i - \mathbf{s}_j \right\| \right), \tag{10}$$

where $\Theta$ is the Heaviside step function, so $\Theta(x) = 1$, for $x \geq 0$, $\Theta(x) = 0$, for $x < 0$, and $\|\cdot\|$ symbolizes the maximum norm of one vector.

Equation (10) is computed for different values of $\varepsilon$, with the vectors obtained from the Equation (4) for each particular case of $m$. If the data obtained, $C_\varepsilon$ versus $\varepsilon$, are arranged in a straight line in a log–log

plot, especially for the intermediate values of $\varepsilon$, that means that the correlation sum follows a power law. For extreme values of $\varepsilon$, the statistical estimation of $C_\varepsilon$ is not reliable, and therefore, the points obtained usually move away from the straight line. Consequently, a straight line, in a log–log plot and for a central region of values of $\varepsilon$, suggests a power law, which in this case, obeys:

$$C_\varepsilon \propto \varepsilon^{\dim_{C_\varepsilon}}, \tag{11}$$

where the exponent $\dim_{C_\varepsilon} \equiv D_2(\varepsilon)$, often a non-integer number, denotes the slope of the line, and it is called the correlation dimension.

For different values of $m$, different values of the slope of the line are generally obtained. The minimum correlation dimension value, for which additional increments of $m$ do not clearly modify its value, implicitly defines the appropriate reconstruction dimension, say $m > 2\lceil D_2 \rceil$, in accordance with Takens's theorem [10], so that the attractor can fully deploy its dynamics (see Figure 8b). If the correlation dimension grows continuously with each value of $m$ initially chosen, the data suggest a random behavior.

False Nearest Neighbors

In a reconstruction space of very low dimension, two points appear to be closer to each other than they are. Two points are considered as real neighbors if the distance between them remains constant as the reconstruction dimension increases. Conversely, the distance between false neighbors continues to increase as long as the reconstruction dimension remains too low.

The underlying principle behind the method is to look for points in the time series that are neighbors in the reconstruction space, but that should not be, as their future time evolution is very different [38,39]. The distances between all points, for different consecutive reconstruction space dimensions, $m$-dimensional and $(m+1)$-dimensional spaces, must be estimated. If the ratio between both distances is higher than a threshold $r$, it says that the neighbors are false neighbors.

If the standard deviation of the data is $\sigma$ and it uses the maximum norm, for computation speed reasons, the percentage of false neighbors $\chi$ amounts to:

$$\chi(r) = \frac{\sum_{n=1}^{N-m-1} \Theta\left(\frac{\left\|\mathbf{s}_n^{(m+1)} - \mathbf{s}_{k(n)}^{(m+1)}\right\|}{\left\|\mathbf{s}_n^{(m)} - \mathbf{s}_{k(n)}^{(m)}\right\|} - r\right) \Theta\left(\frac{\sigma}{r} - \left\|\mathbf{s}_n^{(m)} - \mathbf{s}_{k(n)}^{(m)}\right\|\right)}{\sum_{n=1}^{N-m-1} \Theta\left(\frac{\sigma}{r} - \left\|\mathbf{s}_n^{(m)} - \mathbf{s}_{k(n)}^{(m)}\right\|\right)}, \tag{12}$$

where $\mathbf{s}_{k(n)}^{(m)}$ is the nearest neighbor to $\mathbf{s}_n$ in the $m$-dimensional reconstruction space. The subscript $k(n)$ indexes the time series element, with $k(n) \neq n$, for which $\left\|\mathbf{s}_n - \mathbf{s}_{k(n)}\right\|$ is minimal.

The first term of the numerator in Equation (12), within the summation, is equal to one, if the nearest neighbor is false, that is if the distance increases by a factor greater than $r$ when the reconstruction dimension is increased by one, from $m$ to $m+1$. The second term drops those pairs of points whose initial distance is already greater than $\sigma/r$, since, by definition, they cannot be false neighbors, as on average, the points cannot be further away than $\sigma$. Therefore, in the calculation, these points should not be taken and, thus, appear in the denominator's normalization factor.

The right reconstruction dimension is that for which the percentage of false neighbors falls to approximately zero, as shown in Figure 9. Once it reaches the dimension, it is assumed that the attractor embraces its true spatial configuration, at least from a topological perspective. The results may depend on the chosen delay $\tau$. In any case, if it seeks to be able to differentiate the chaos from the noise, it is essential to validate the results using surrogate data testing [29].

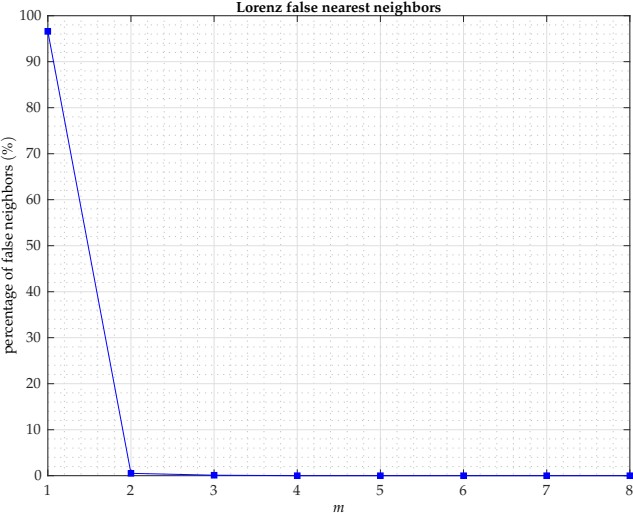

**Figure 9.** From the dynamic variable $x$ of the Lorenz attractor: false nearest neighbors. From $m = 3$, the number of false nearest neighbors is almost zero.

## 3. Results

The study methodology of the PPG signal to finally get the phase space reconstruction and draw interesting conclusions about the underlying physiological dynamics followed the next steps: first, we took 15,000 points, which corresponded to one minute of recorded signal. Obviously, the selected epoch was assumed stationary once the procedure, described in Section 2.2.2, was applied to the data. Then, we studied its graphical representation, considering different options for different graphic representations, as shown in Figures A1–A3 (see Appendix A), to sustain the presence of a latent dynamic structure.

Second, we tried to figure out the attractor more akin to the original, based on the optimal lag $\tau$ (see Section 2.3.1), according to the autocorrelation function (AF) (Figure 10a) and the mutual information (MI) (Figure 10b), as well as on the optimal embedding dimension $m$ (see Section 2.3.2), in accordance with the principal component analysis (PCA) (Figure 11a), the correlation dimension (D2) (Figure 11b), and the false nearest neighbors method (FNN) (Figure 11c).

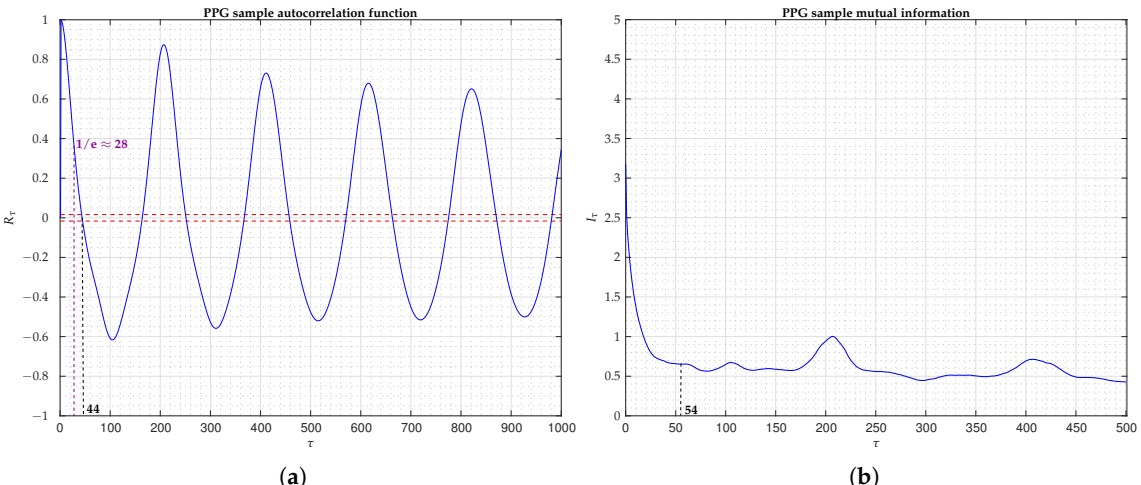

(**a**)          (**b**)

**Figure 10.** From a sample PPG signal (Subject Number 4): (**a**) autocorrelation function (AF), where the first zero crossing is when $\tau = 44$; and (**b**) mutual information (MI), where the first minimum is with $\tau = 54$.

We applied the methodology to study 10 min of the PPG signal from each subject, and the results were similar to the ones shown in this paper. We saw in the Lorenz flow, according to the AF, that the criterion of $1/e$ was approximately similar to the MI result because the attractor was chaotic and both linear and nonlinear correlations decayed rapidly, as illustrated in Figure 7. In the PPG signal, the criterion of the first zero crossing was closer to the MI result (see Figure 10) because the PPG signal was predominantly quasi-periodic on small timescales [22], while the correlations were stronger. Either way, the criterion of the first minimum of MI was adopted because it ensured statistical independence between the values, in linear and nonlinear terms.

The parameters $\tau$ and $m$ calculated for five subjects, with results from each of the methods previously described, are in Table 1. The optimal $\tau$ depended on the subject considered and on the time interval in which it estimated. Probably the range of optimal $\tau$ values varied with each subject and could be a hallmark of each individual. It should make a careful study with all subjects and under changing conditions of the organism in its psychosomatic relationship with the surrounding environment.

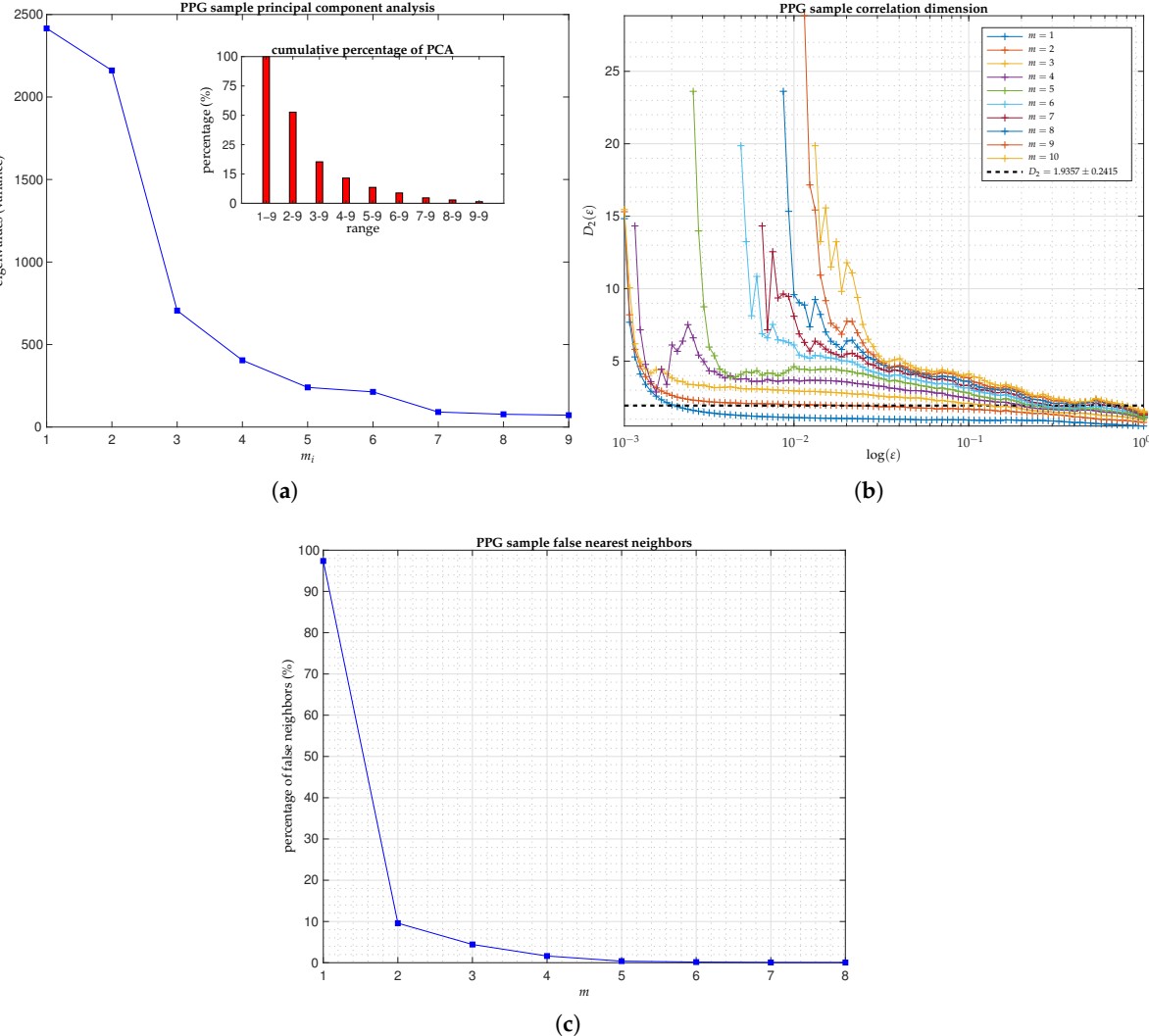

**Figure 11.** From a sample PPG signal (Subject Number 4): (**a**) Principal component analysis (PCA). The inner box clearly shows how the dynamic behavior of the attractor can be characterized with roughly seven directions. (**b**) Correlation dimension ($D_2$). It can be seen how from $m = 4$, the correlation dimension was confined to a constant value in an intermediate and reduced range of scales $\varepsilon$. (**c**) False nearest neighbors. From $m = 5$, the number of false nearest neighbors is almost zero.

**Table 1.** Parameters $\tau$ and $m$ of the PPG signals, acquired from five individuals chosen at random, for the different methods already explained, namely autocorrelation function (AF), mutual information (MI), principal component analysis (PCA), correlation dimension ($D_2$), and false nearest neighbors (FNN). The parameters are calculated for all the subjects of the experimental research (40 PPG signals), proving that with the FNN method, all of them showed $m = 5$.

| Evaluated Signal | $\tau$ | | | $m$ | |
|---|---|---|---|---|---|
| | AF | MI | PCA | $D_2$ | FNN |
| Subject Number 1 (PPG1) | 52 | 81 | 5 | 9 | 5 |
| Subject Number 2 (PPG2) | 37 | 35 | 6 | 5 | 5 |
| Subject Number 3 (PPG3) | 29 | 30 | 5 | 5 | 5 |
| Subject Number 4 (PPG4) | 44 | 54 | 6 | 5 | 5 |
| Subject Number 5 (PPG5) | 44 | 33 | 6 | 5 | 5 |

Once we had the parameter $\tau$ for each subject, calculated by the MI criterion, we proceeded to the calculation of parameter $m$. Figure 11 visually depicts the results for a sample PPG signal, which corresponded to Subject Number 4 in Table 1. We saw that multivariate analysis, such as the PCA linear method, did not provide the ideal value of $m$, although it was not unreasonable, mostly because physical systems do not usually have very high dimensions. The correlation dimension gave the value of $m > 2 \lceil D_2 \rceil$, which except for the first subject, amounted to a value of $m = 5$. For the FNN method, all calculations agreed with $m = 5$. This later method was a good estimator of $m$ because in resorting to topological aspects of the reconstructed state space, the effect of noise on algorithmic computation was minimal. Finally, with the selected parameters $\tau$ and $m$, the determinism present in the state space reconstruction was corroborated by the index of smoothness (see Section 2.2.2), obtaining a relatively low value compared to the surrogate data. In addition, a statistical hypothesis test showed that the trajectory followed by the state vectors did not respond to a linear stochastic process, according to the surrogate data, but rather a nonlinear deterministic process.

We concluded that all individuals had the same embedding dimension $m = 5$, although we thought that it may vary depending on the subject's physical and psychological states. Consequently, the five first-order differential equations' system described the dynamic system that generated a typical PPG signal from a healthy young individual, in which various parameters could contribute to the mechanism of self-regulation of the underlying physiological process. Figures A4–A8 (see Appendix B) show how the attractors of the subjects were different for the calculated values of $\tau$; in the accompanying video clips (Video S1), we could watch the evolution of the attractor geometry for different lags, from one to 150, for both 2D and 3D phase diagrams. Ongoing studies on the effect of changes in the parameters $\tau$ and $m$ on the dynamics of the underlying physiological process will shed more light in this regard.

## 4. Discussion

The well-known phase space reconstruction served as a starting point for the analysis or modeling of the dynamics of physiological systems reconstructed from a single biological signal. The dynamics that described a PPG signal was deterministic. The embedding dimension $m$, for healthy young subjects, was five, regardless of the subject and the time interval estimation. The optimal lag or delay time $\tau_{opt}$ depended on the subject and calculated time interval. A further study, with individuals of different ages and with different proven pathologies, will be carried out to confirm if the embedding dimension remains the same. In addition, it will be thoroughly examined if the variation of $\tau$, even of $m$, in each individual is an indicator of the mood and physical status in which he or she is, either as a result of a sporadic situation, such as an episode of stress, a more severe disease, or the age related evidence for biological and physiological decline. We described, in some detail, the most usual and clear methodology to calculate the phase space reconstruction because we found that in its application to biological signals, it has not been well understood at the physiological level, and its discriminant

potential in the clinical setting could not be sufficiently exploited. In this sense, its effectiveness could be corroborated with the most modern state space reconstruction techniques that are less heuristic and with a more consolidated mathematical formalism [40].

**Supplementary Materials:** The following are available online at http://www.mdpi.com/2076-3417/10/4/1430/s1, Video S1: 2D-3D phase diagrams PPG signals.

**Author Contributions:** Conceptualization, J.d.P.-C. and A.P.G.-M.; methodology, J.d.P.-C.; software, J.d.P.-C.; validation, J.d.P.-C., D.F.-J., and A.M.U.; formal analysis, J.d.P.-C., D.F.-J., and A.M.U.; data curation, A.M.U.; writing, original draft preparation, J.d.P.-C.; writing, review and editing, J.d.P.-C. and A.P.G.-M.; visualization, J.d.P.-C.; supervision, A.P.G.-M.; project administration, A.M.U.; funding acquisition, A.P.G.-M. All authors have read and agreed to the published version of the manuscript.

**Funding:** This research received no external funding.

**Acknowledgments:** The authors would like to thank Life Supporting Technologies Group (LST-UPM) for taking part in Project FIS-PI12/00514, from MINECO. Furthermore, they want to thank especially the support given by Raúl Durán Díaz, from the University of Alcalá (UAH) and the Spanish National Research Council (CSIC), providing a cluster of servers with which to execute some of the computational algorithms of this work. The cluster consists of ten computing nodes, with a total of 64 cores, an Intel Xeon architecture, and 166 GB of memory.

**Conflicts of Interest:** The authors declare no conflict of interest.

## Appendix A. Different Methods of Graphical Representation of a Sample PPG Signal

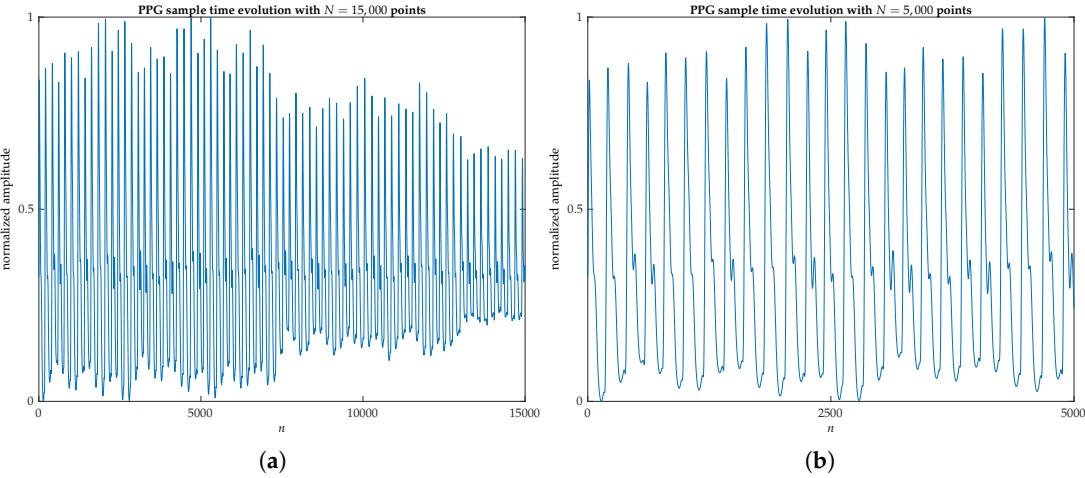

(**a**)          (**b**)

**Figure A1.** From a sample PPG signal: (**a**) one minute PPG recorded signal of a healthy young subject; (**b**) the first 20 s of (**a**).

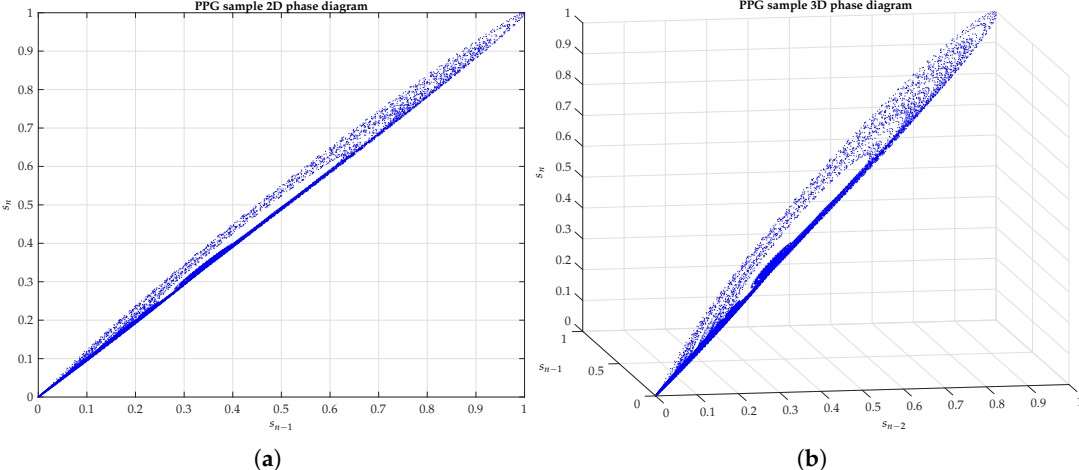

(**a**)          (**b**)

**Figure A2.** *Cont.*

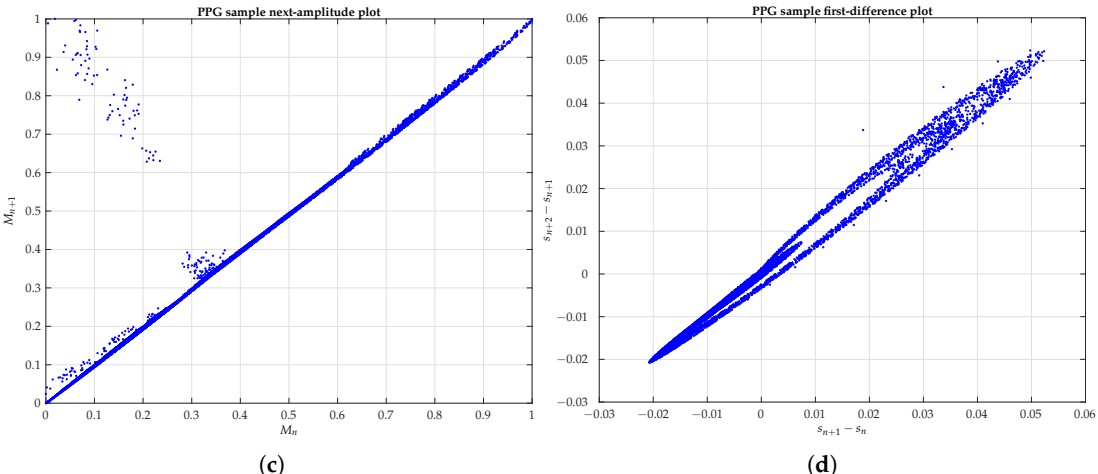

(**c**)    (**d**)

**Figure A2.** From a sample PPG signal and for $\tau = 1$: (**a**) 2D phase diagram; (**b**) 3D phase diagram; (**c**) successive-maxima plot; (**d**) difference plot.

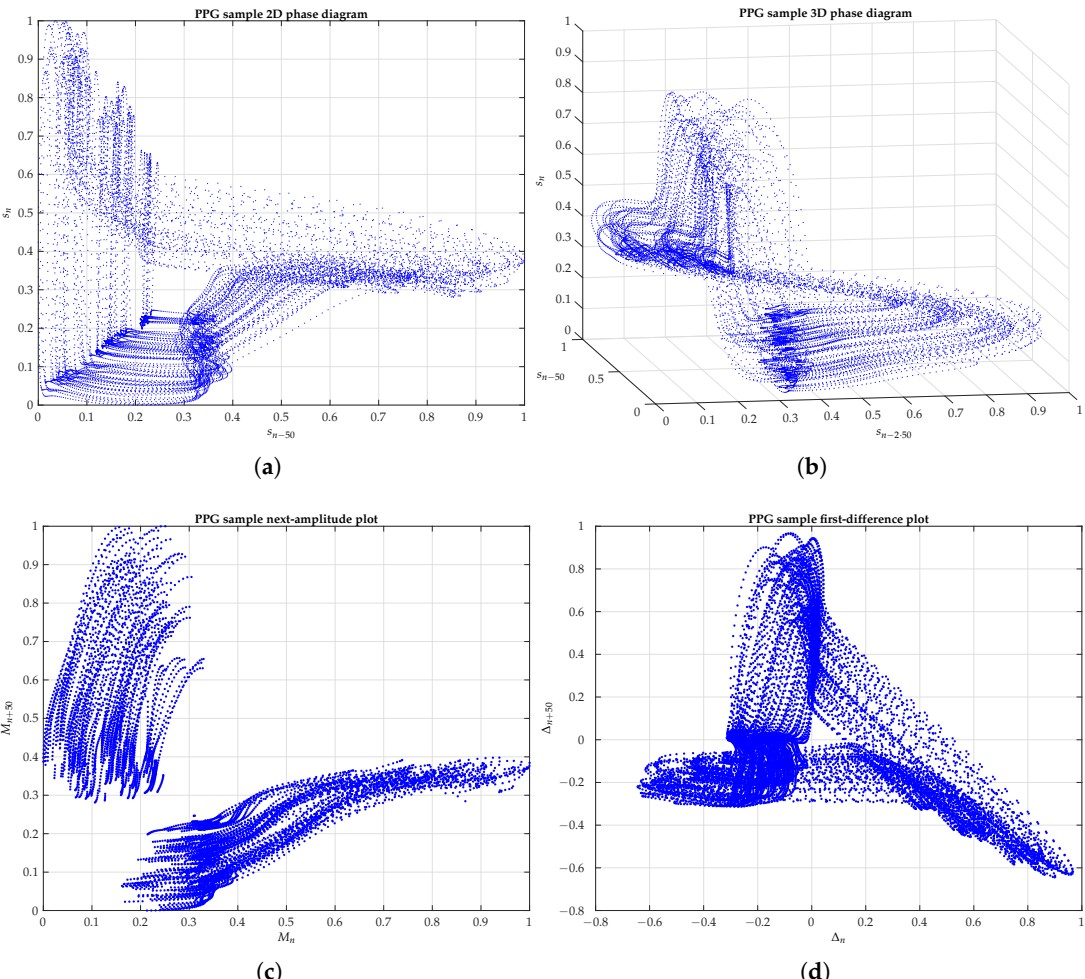

(**a**)    (**b**)

(**c**)    (**d**)

**Figure A3.** From a sample PPG signal and for $\tau = 50$: (**a**) 2D phase diagram; (**b**) 3D phase diagram; (**c**) successive-maxima plot; (**d**) difference plot.

## Appendix B. 2D and 3D Phase Diagrams of Five PPG Signals Randomly Chosen

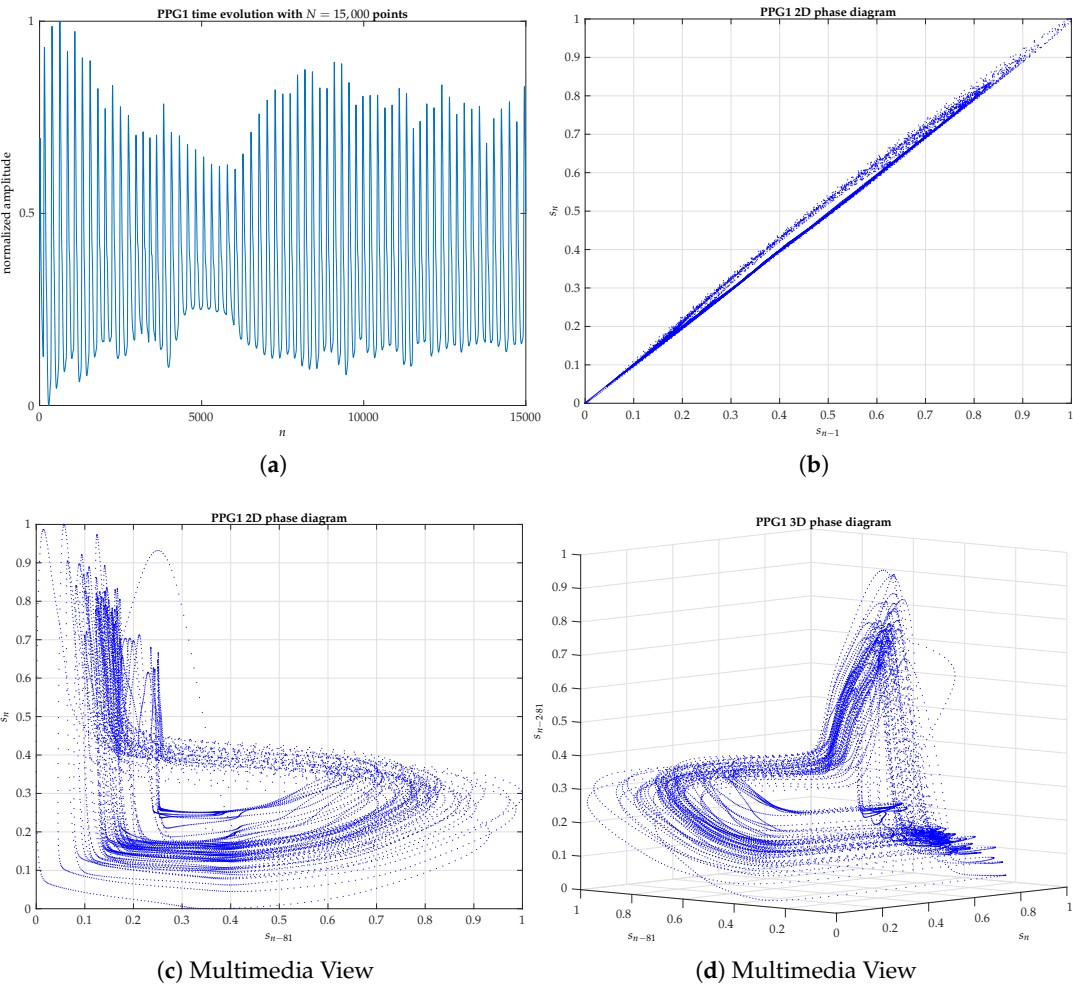

(**c**) Multimedia View        (**d**) Multimedia View

**Figure A4.** PPG signal of the healthy young Subject Number 1 (PPG1): (**a**) One minute PPG recorded signal. (**b**) PPG1 2D phase diagram with $\tau = 1$. (**c**) PPG1 2D phase diagram with $\tau = 81$, according to Table 1. PPG1 video clip of the evolution of phase diagram from $\tau = 1$ to $\tau = 150$ in 2D. (**d**) PPG1 3D phase diagram with $\tau = 81$. PPG1 video clip of the evolution of phase diagram from $\tau = 1$ to $\tau = 150$ in 3D.

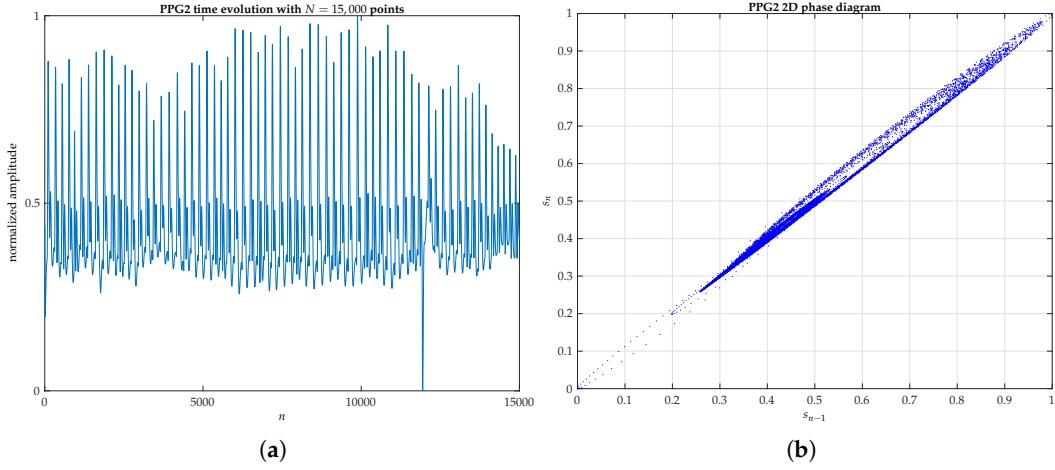

(**a**)        (**b**)

**Figure A5.** *Cont.*

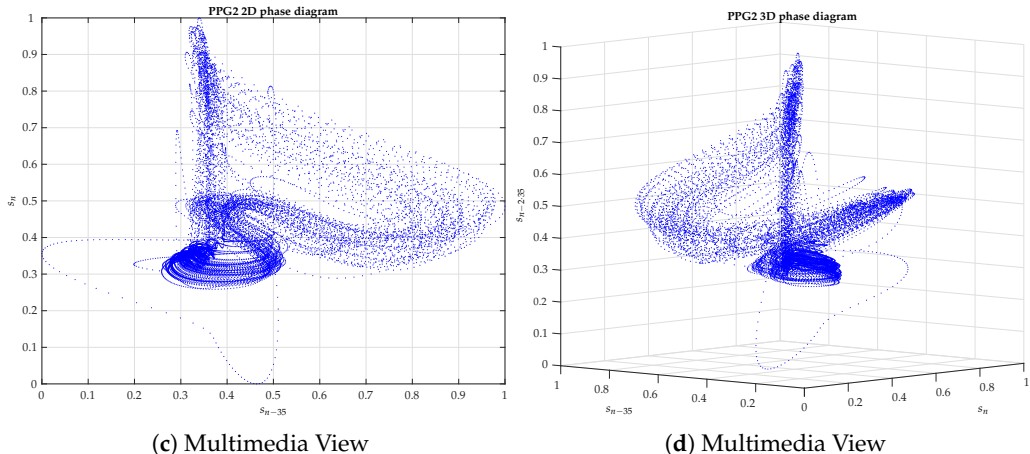

(**c**) Multimedia View                    (**d**) Multimedia View

**Figure A5.** PPG signal of the healthy young Subject Number 2 (PPG2): (**a**) One minute PPG recorded signal. (**b**) PPG2 2D phase diagram with $\tau = 1$. (**c**) PPG2 2D phase diagram with $\tau = 35$, according to Table 1. PPG2 video clip of the evolution of phase diagram from $\tau = 1$ to $\tau = 150$ in 2D. (**d**) PPG2 3D phase diagram with $\tau = 35$. PPG2 video clip of the evolution of phase diagram from $\tau = 1$ to $\tau = 150$ in 3D.

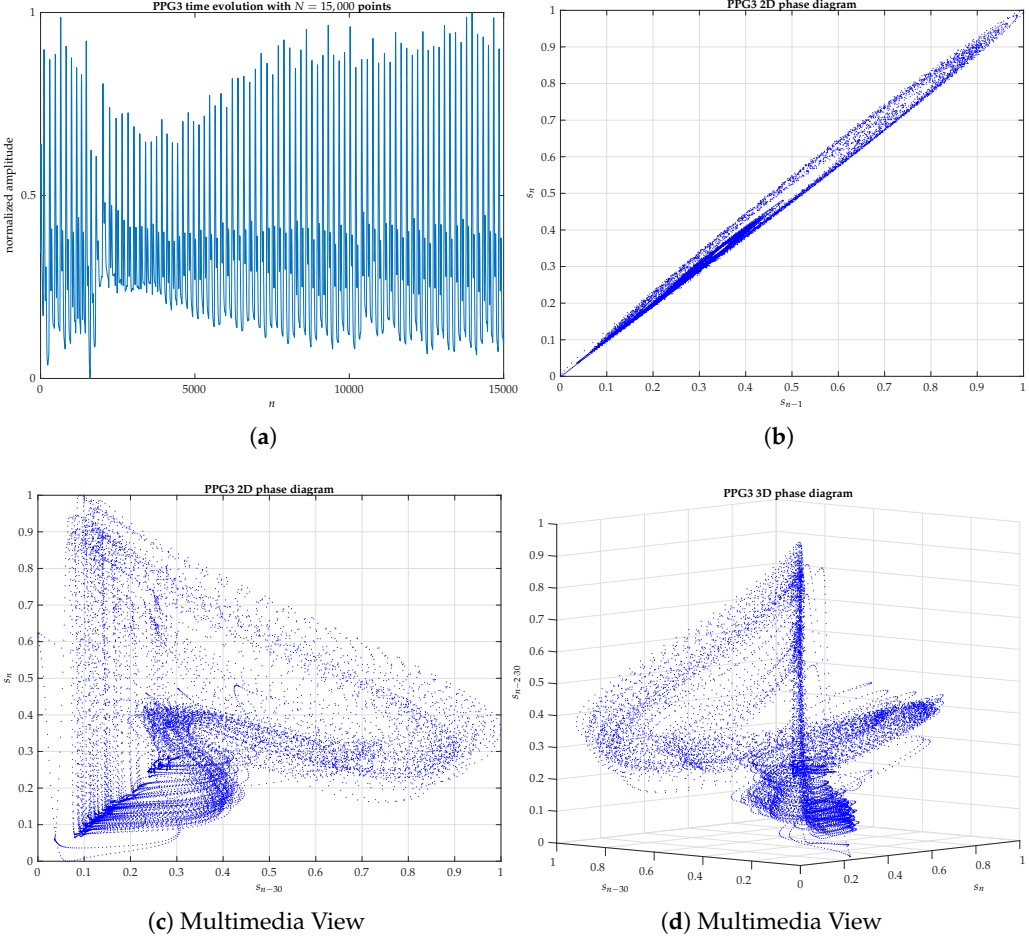

(**c**) Multimedia View                    (**d**) Multimedia View

**Figure A6.** PPG signal of the healthy young Subject Number 3 (PPG3): (**a**) One minute PPG recorded signal. (**b**) PPG3 2D phase diagram with $\tau = 1$. (**c**) PPG3 2D phase diagram with $\tau = 30$, according to Table 1. PPG3 video clip of the evolution of phase diagram from $\tau = 1$ to $\tau = 150$ in 2D. (**d**) PPG3 3D phase diagram with $\tau = 30$. PPG3 video clip of the evolution of phase diagram from $\tau = 1$ to $\tau = 150$ in 3D.

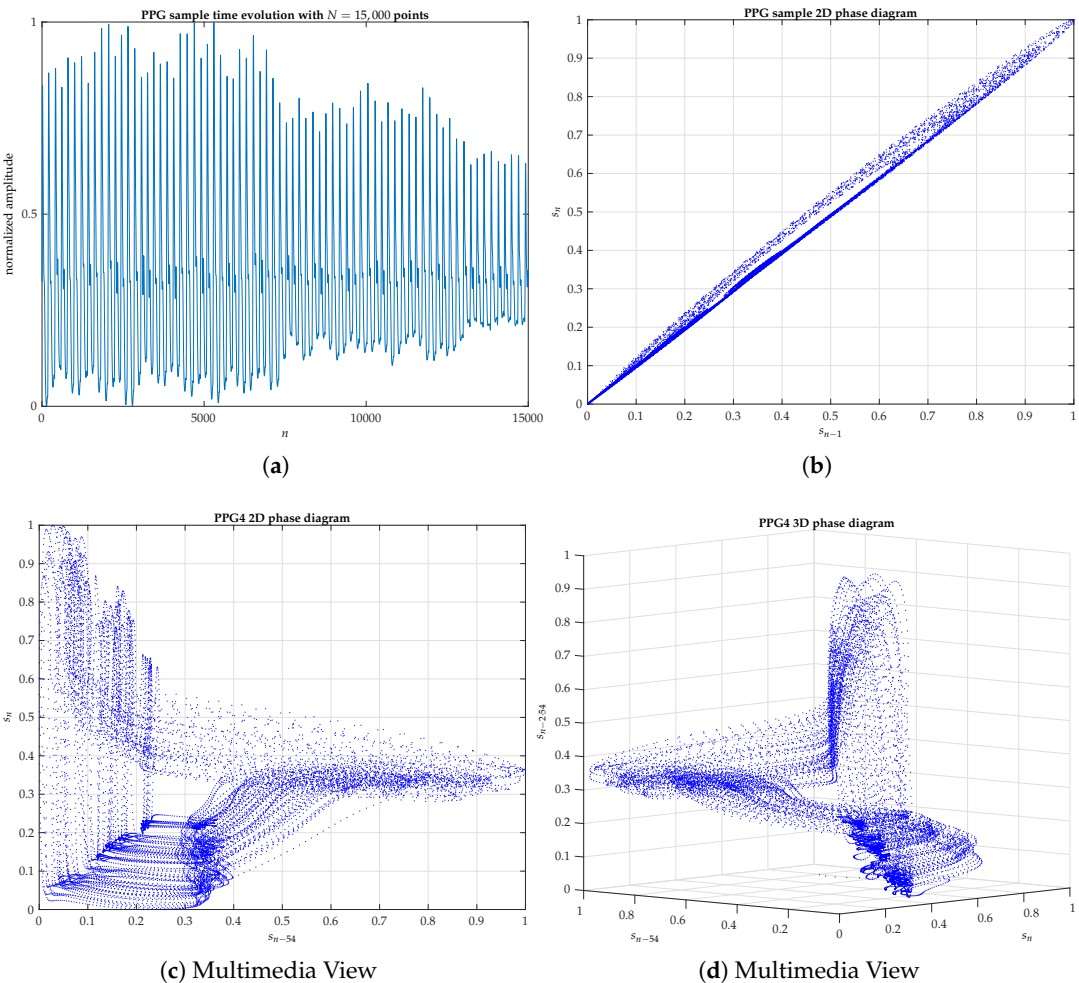

(**c**) Multimedia View    (**d**) Multimedia View

**Figure A7.** PPG signal of the healthy young Subject Number 4 (PPG4): (**a**) One minute PPG recorded signal. (**b**) PPG4 2D phase diagram with $\tau = 1$. (**c**) PPG4 2D phase diagram with $\tau = 54$, according to Table 1. PPG4 video clip of the evolution of phase diagram from $\tau = 1$ to $\tau = 150$ in 2D. (**d**) PPG4 3D phase diagram with $\tau = 54$. PPG4 video clip of the evolution of phase diagram from $\tau = 1$ to $\tau = 150$ in 3D.

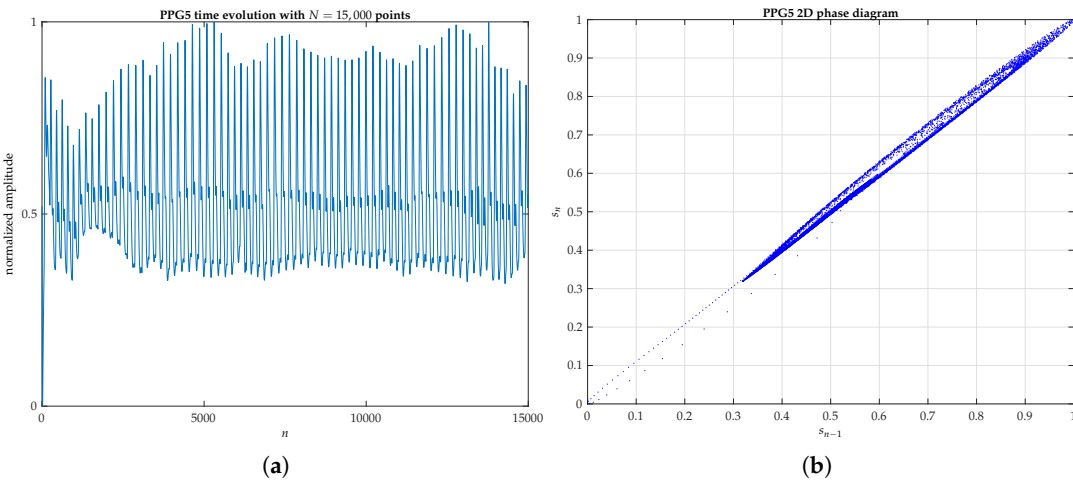

(**a**)    (**b**)

**Figure A8.** *Cont.*

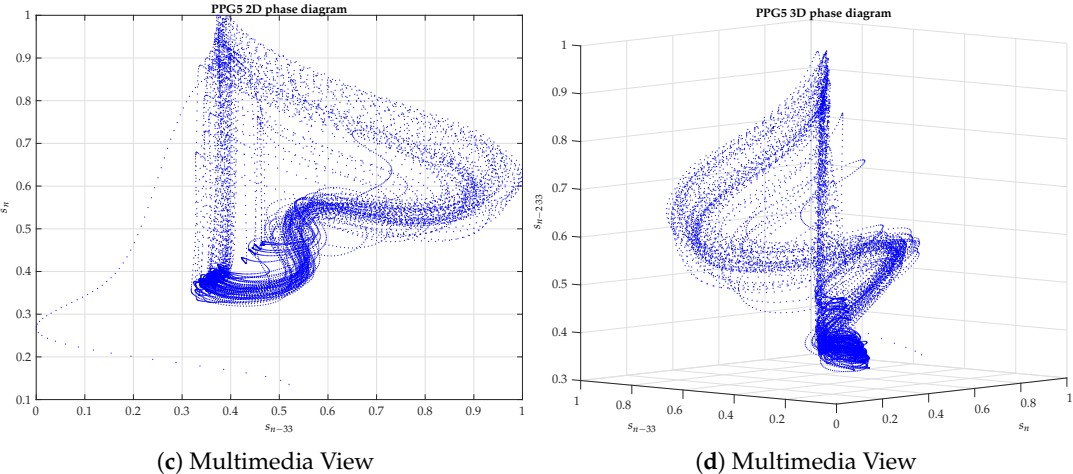

(**c**) Multimedia View  (**d**) Multimedia View

**Figure A8.** PPG signal of the healthy young Subject Number 5 (PPG5): (**a**) One minute PPG recorded signal. (**b**) PPG5 2D phase diagram with $\tau = 1$. (**c**) PPG5 2D phase diagram with $\tau = 33$, according to Table 1. PPG5 video clip of the evolution of phase diagram from $\tau = 1$ to $\tau = 150$ in 2D. (**d**) PPG5 3D phase diagram with $\tau = 33$. PPG5 video clip of the evolution of phase diagram from $\tau = 1$ to $\tau = 150$ in 3D.

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
