# Peer review of "Phase Space Reconstruction from a Biological Time Series: A Photoplethysmographic Signal Case Study"

_applsci, doi:10.3390/app10041430_

Round 1
Reviewer 1 Report
In this manuscript, the phase space reconstruction from one scalar time series is proposed and applied to the PPG signal. It is an interesting topic for readers. However, the description of the proposed approaches were insufficient. The importance and the motivation are not detailed in the first paragraph of the section of introduction. The method should be systematically described. The authors also need to provide the description of the system block diagram (or the flowchart) to help readers understand the system process. In the results, only five subjects were examined. Can the author provide more subjects for objectively evaluating the proposed approaches? Moreover, the proposed approach was examined and the authors need to provide other approach for objectively comparison.
Reviewer 2 Report
The same paper appears here : https://arxiv.org/pdf/1910.05410.pdf
I suspect some kind of autoplagiarism.
My reference will be stopped until authors explanation.
Author Response
Response to Reviewer 2
Comments
Point 1: The same paper appears here : https://arxiv.org/pdf/1910.05410.pdf I suspect some kind of autoplagiarism. My reference will be stopped until authors explanation.
Response 2: We feel sorry about your impression.
As it say in https://www.editage.com/insights/can-you-explain-what-is-an-arxiv-publication
“arXiv is not a journal; therefore articles on arXiv cannot be regarded as publications per se. Typically, papers deposited on this repository are pre-publication versions and authors can add a DOI to their pre-print version on arXiv after it has been published. Also, the files uploaded on arXiv could have different copyright statuses.”
We comment on the cover letter to the editor that we have this pre-print. Also the MDPI editorial has his own Preprints plataform, that we did not know.
Reviewer 3 Report
Authors showed that the phase space reconstruction at the physiological level, and its discriminant potential in the clinical setting no sufficiently exploit.
My only concern is an unclear hypothesis and conclusion. Authors should clearly show the hypothesis and conclusion in the text and abstract.
Please clarify the detail of your cohort (40 students, between 18 and 30 years old). Gender, Comorbidities, Body mass index, etc.
Round 2
Reviewer 2 Report
In article new modalities to observe signals are presented. I agree it is an interresting area of investigation, but in my opinion there is a lack of serious conclussions. Much research work must be done in order to show, how to find abnormalities in recorded signals. How to categorize them to a disease or measurement artifact.